# RatXcan: A framework for cross-species integration of genome-wide association and gene expression data

Natasha Santhanam[1☯], Sandra Sanchez-Roige[2,3,4☯], Sabrina Mi[2], Yanyu Liang[1], Apurva S. Chitre[2], Daniel Munro[2], Denghui Chen[2], Jianjun Gao[2], Angel Garcia-Martinez[5], Anthony M. George[6], Alexander F. Gileta[2], Wenyan Han[5], Katie Holl[7], Alesa Hughson[8], Christopher P. King[9], Alexander C. Lamparelli[9], Connor D. Martin[6], Festus Nyasimi[1], Celine L. St. Pierre[2], Sarah Sumner[1], Jordan Tripi[9], Tengfei Wang[5], Hao Chen[5], Shelly Flagel[8], Keita Ishiwari[6,10], Paul Meyer[6,9], Oksana Polesskaya[2], Laura Saba[11], Leah C. Solberg Woods[12], Abraham A. Palmer[2,3*], Hae Kyung Im[1*]

1 Department of Medicine, Section of Genetic Medicine, The University of Chicago, Chicago, Illinois, United States of America, 2 Department of Psychiatry, University of California San Diego, La Jolla, California, United States of America, 3 Institute for Genomic Medicine, University of California San Diego, La Jolla, California, United States of America, 4 Department of Medicine, Division of Genetic Medicine, Vanderbilt University Medical Center, Nashville, Tennessee, United States of America, 5 University of Tennessee Health Science Center, Department of Pharmacology, Addiction Science and Toxicology, Memphis, Tennessee, United States of America, 6 University at Buffalo, Clinical and Research Institute on Addictions, University at Buffalo, Buffalo, New York, United States of America, 7 Medical College of Wisconsin, Department of Pediatrics, Milwaukee, Wisconsin, United States of America, 8 University of Michigan, Department of Psychiatry, Ann Arbor, Michigan, United States of America, 9 University at Buffalo, Department of Psychology, Buffalo, New York, United States of America, 10 University at Buffalo, Department of Pharmacology and Toxicology, Buffalo, New York, United States of America, 11 University of Colorado Anschutz Medical Campus, Department of Pharmaceutical Sciences, Aurora, Colorado, United States of America, 12 Wake Forest University School of Medicine, Department of Internal Medicine, Winston-Salem, North Carolina, United States of America

☯ These authors contributed equally to this work.
* aap@ucsd.edu (AAP); haky@uchicago.edu (HKI)

## Abstract

Genome-wide association studies (**GWAS**) have implicated specific alleles and genes as risk factors for numerous complex traits. However, translating GWAS results into biologically and therapeutically meaningful discoveries remains extremely challenging. Most GWAS results identify noncoding regions of the genome, suggesting that differences in gene regulation are the major driver of trait variability. To better integrate GWAS results with gene regulatory polymorphisms, we previously developed PrediXcan (also known as "transcriptome-wide association studies" or **TWAS**), which maps SNPs to predicted gene expression using GWAS data. In this study, we developed RatXcan, a framework that extends this methodology to outbred heterogeneous stock (**HS**) rats. RatXcan accounts for the close familial relationships among HS rats by modeling the relatedness with a random effect that encodes the genetic relatedness. RatXcan also corrects for polygenic-driven inflation because of the equivalence between a relatedness random effect and the infinitesimal polygenic model. To develop RatXcan, we trained transcript predictors for 8,934 genes using reference genotype and expression data from five rat brain regions. We found

**Data availability statement:** All data for this paper are available in https://doi.org/10.5281/zenodo.13996957 All code for this paper are available in https://doi.org/10.5281/zenodo.13997008 The software implementation is publicly available on a GitHub repository (https://github.com/hakyimlab/rat_genomics_paper_pipeline_2024). Prediction models for gene expression in all five brain tissues in rats are also available at https://predictdb.org. Association results are attached as supplementary tables and also available in the RatXcan portal (http://imlab.shinyapps.io/RatXcan).

**Funding:** This work was supported by the National Institute on Alcohol Abuse and Alcoholism (R01AA029688 to AAP, HKI, SSR), the National Institute on Drug Abuse (NIDA DP1DA054394 to SSR), the National Institute of Diabetes and Digestive and Kidney Diseases (P30DK020595 to HKI), and the National Cancer Institute to (3R01CA242929-04S1 to HKI). The rat datasets used were supported by NIH NIDA (P50DA037844 and R01AA029688 to AAP). The funders had no role in study design, data collection and analysis, decision to publish, or preparation of the manuscript.

**Competing interests:** The authors have declared that no competing interests exist.

that the cis genetic architecture of gene expression in both rats and humans was sparse and similar across brain tissues. We tested the association between predicted expression in rats and two example traits (body length and BMI) using phenotype and genotype data from 5,401 densely genotyped HS rats and identified a significant enrichment between the genes associated with rat and human body length and BMI. Thus, RatXcan represents a valuable tool for identifying the relationship between gene expression and phenotypes across species and paves the way to explore shared biological mechanisms of complex traits.

## Author summary

Understanding how genetic variation affects phenotypic variation is critical to leveraging the wealth of genetic studies to make biologically and therapeutically useful discoveries. Since most of the genetic loci associated with complex diseases are regulatory in nature—meaning that they do not alter protein coding but rather subtly affect gene expression—transcriptome-wide association studies have been developed. However, these apply only to human data where large samples of unrelated individuals are available. For animal models, relatedness is much higher, causing higher false-positive rates. We propose a computationally efficient method to address this problem and find shared biology between humans and rats. Taken together, our development paves the way to further explore shared biological mechanisms of complex traits across species.

## Introduction

Over the last decade, genome-wide association studies (**GWAS**) have identified numerous genetic loci that contribute to biomedically important traits [1]. GWAS have demonstrated that most traits have a highly polygenic architecture, meaning that numerous genetic variants with individually small effects confer risk [2].

However, translating these results into biologically meaningful discoveries remains extremely challenging [3–5]. One major challenge is that ~90% of the GWAS loci implicate noncoding regions; these presumably regulatory loci cannot be confidently ascribed to the nearest gene. To address this challenge, we previously developed PrediXcan [6], the first of a class of methods known as transcriptome-wide association studies [**TWAS**; [6,7]] which seek to identify causal genes by testing the role of gene expression traits on phenotypic variation. This is accomplished by correlating the genetically predicted expression of genes with the phenotype of interest.

Model organisms can complement human GWAS findings by providing a platform to experimentally test or perturb biological mechanisms impacted by genetic variation in the context of specific behaviors, tissues, and molecular systems. The methodology for GWAS in non-human organisms has been successful [8–10]. However, whether or not the genetic architecture of complex traits of model organisms accurately mirrors that of humans remains controversial [11–13].

In this study, we developed RatXcan to extend the PrediXcan methodology to outbred heterogeneous stock (**HS**) rats. RatXcan is predicated on the regulatory nature of most GWAS loci [14] and uses gene expression to nominate causal genes for complex traits. We selected HS rats because they are a well-characterized outbred mammalian population for which dense genotype, phenotype, and gene expression data are available in thousands of subjects

[15]. In the development of RatXcan, we accounted for the higher degree of familial relatedness observed in laboratory bred-colonies like HS rats and polygenicity-driven inflation [16] implementing a computationally efficient mixed effects modeling. The utility of this mixed effects modeling goes beyond the rat data presented here and should be applicable to other species data as well as account for population structure in human data. Finally, using this methodology, we explored whether similar complex traits across species, namely height/body length and BMI, are influenced by regulatory polymorphisms in orthologous genes by applying TWAS to rats and humans. Thus, we demonstrated that RatXcan can be effectively employed to test the conservation of gene–phenotype relationships between species.

## Results

### Experimental setup

To build a framework for translating genetic results between species, we trained gene expression models as follows. In the *training stage*, we investigated the genetic architecture of gene expression in rats and built prediction models of gene expression using genotype and transcriptome data from five brain regions sampled from 88 HS rats [17]. In the *association stage*, we used genotype data and models from the training stage to predict the transcriptome in a non-overlapping *target set* of 5,401 rats that had been used in two prior GWAS for body length and BMI [8,18]. We tested for associations between the genetically predicted gene expression and body length and BMI by extending the PrediXcan framework—which was originally developed for use in humans [6]—to account for the higher relatedness in rats ('RatXcan'). We did this by using a random effect that encodes both the genetic relatedness and a fully polygenic trait (infinitesimal model; see Methods). Thus, RatXcan corrects both for relatedness and the polygenicity-driven inflation reported recently by [16]. Finally, we examined the overlap of rat trait-associated genes with human results from the PhenomeXcan database [19].

### Genetic architecture of gene expression across brain tissues

To inform the optimal prediction model training, we examined the genetic architecture of gene expression in HS rats by quantifying heritability and polygenicity across five brain tissues. Because the results for each tissue are similar, we only summarize results for one of the five tissues (nucleus accumbens core or **NAcc**); the remaining tissues are reported in S1 Fig.

We calculated the heritability of expression for each gene by estimating the proportion of variance explained (**PVE**) using a Bayesian Sparse Linear Mixed Model (**BSLMM**) [20]. We restricted the feature set to variants within 1 Mb upstream of the transcription start and 1 Mb downstream of the transcription end of each gene since this is expected to capture most cis-eQTLs in this population, similar to our prior work in [17]. Among the 15,216 genes considered, 3,438 genes had a 95% credible set lower boundary >1%) in the NAcc (Figs 1A and S1 for remaining tissues). The mean local heritability (± 1 Mb) ranged from 13.5% to 15.5% for all brain tissues tested (Table 1). We identified a similar heritability distribution in humans (Figs 1C and S2) based on whole blood samples from GTEx.

Next, to evaluate the polygenicity of gene expression levels, we examined whether predictors with more polygenic or sparse architecture correlated better with observed expression. We fitted elastic net regression models using a range of mixing parameters from 0 to 1 (Fig 1B). The leftmost parameter value of 0 corresponds to ridge regression, which is fully polygenic and uses all cis-variants. Larger values of the mixing parameters yield more sparse predictors, meaning that the number of variants used decreases as the mixing parameter increases. The rightmost value of 1 corresponds to lasso regression, which yields the most sparse predictor within the elastic net family. We did not use a linear mixed model (**LMM**)

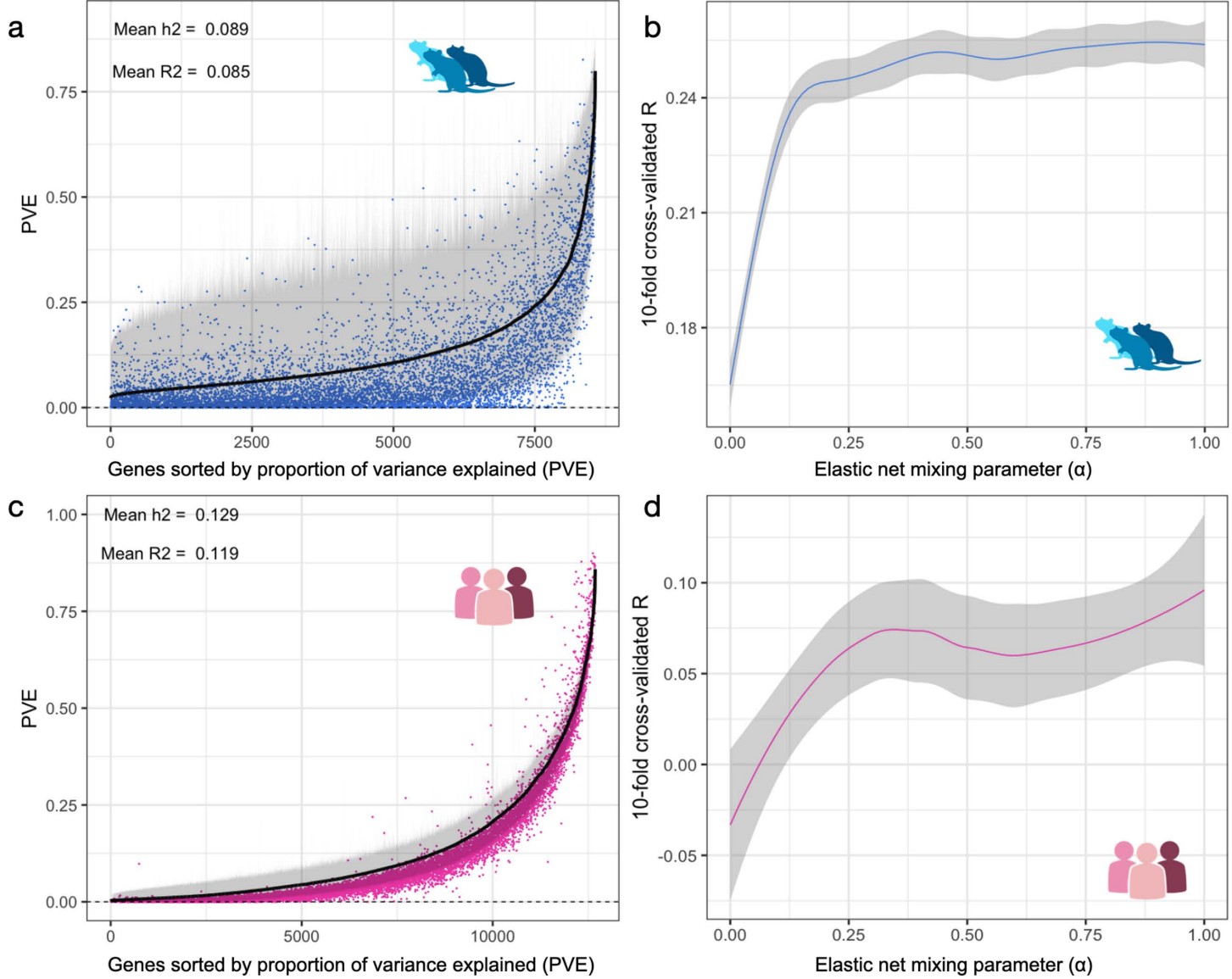

**Fig 1. Heritability and sparsity of gene expression in both rats and humans.** a) cis-heritability of gene expression levels in the NAcc of rats calculated using BSLMM (black). We show only rat genes (N = 10,268) that have an ortholog in the human GTEx data. On the x-axis, genes are ordered by their heritability estimates. 95% credible sets are shown in gray for each gene. Blue dots indicate the prediction performance (cross validated $R^2$ between predicted and observed expression). b) Cross-validated prediction performance in rats (Pearson correlation $R$) as a function of the elastic net parameter ranging from 0 to 1. c) cis-heritability of gene expression levels in whole blood tissue in humans from GTEx. We show only the same 10,268 orthologous genes shown for rats. On the x-axis, genes are ordered by their heritability estimates. 95% credible sets are shown in gray for each gene. Pink dots indicate the prediction performance (cross validated $R^2$ between predicted and observed expression). d) Cross-validated prediction performance in humans (Pearson correlation $R$) as a function of the elastic net parameter ranging from 0 to 1.

because in a prior publication [17] we demonstrated that there was essentially no difference between using a linear model and an LMM for mapping cis-eQTLs in this dataset, perhaps in part because the rats were chosen to be distantly related (e.g., no siblings).

We used the 10-fold cross-validated Pearson correlation ($R$) between predicted and observed values as a measure of performance (Spearman correlation yielded similar results). We observed a substantial drop in performance towards the more polygenic end of the mixing parameter spectrum (Fig 1B). We observed similar results using human gene expression data

**Table 1. Summary of heritability and prediction performance in rats.** The table shows the number of rats used in the prediction, number of genes predicted per model ($R^2>0.01$), the average prediction performance $R^2$ (after filtering $R^2<0.01$), and average cis-heritability (cis $h^2$), for all gene transcripts.

| Brain Region | # Rats | # Genes Predicted | Average $R^2$ | Average cis $h^2$ |
|---|---|---|---|---|
| Nucleus Accumbens Core (NAcc) | 78 | 5,879 | 11.7% | 15.3% |
| Infralimbic Cortex (IL) | 83 | 5,927 | 11.6% | 15.2% |
| Lateral Habenula (LHb) | 83 | 5,957 | 11.6% | 13.5% |
| Prelimbic Cortex (PL) | 81 | 5,947 | 11.5% | 15.5% |
| Orbitofrontal Cortex (OFC) | 82 | 5,891 | 11.7% | 15.0% |

from whole blood samples in GTEx individuals (Fig 1D). Overall, these results indicate that the cis component of the genetic architecture of gene expression in HS rats is sparse, similar to that of humans [21].

## Generation of prediction models of gene expression in rats

We trained elastic net predictors for all genes in all five brain regions. Based on the relative performance across different elastic net mixing parameters, we chose a parameter value of 0.5, which yielded slightly less sparse predictors than lasso but provided robustness against missing or low-quality variants; this is the same value that we have used with humans datasets [6]. The procedure yielded 5,879-5,957 genes across five brain tissues from the available 14,908–15,130 genes after QC (Table 1). The 10-fold cross-validated prediction performance ($R^2$) ranged from 0 to 80%; after filtering out genes with $R^2<0.01$, the mean $R^2$ was 11.7% in the NAcc). As shown in Table 1, mean prediction $R^2$ was consistently lower than mean heritability for all tissues. Prediction performance values followed the heritability curve, confirming that in both rats and humans genes with highly heritable expression tend to be better predicted than genes with low heritability (Fig 1A and 1B).

In Fig 2A and 2B, we show the prediction performance of the best predicted genes in HS rats (*Mgmt*, $R^2 = 0.72$) and humans (*RPS26*, $R^2 = 0.74$). Across all genes, we found that the prediction performance in HS rats was positively correlated with that of humans (*R = 0.061, P* = 8.03 x $10^{-6}$; Fig 2C). Furthermore, genes that were well-predicted in one tissue were also well-predicted in another tissue (Fig 2D and 2E). The correlation of prediction performance across various brain regions ranged from 58 to 84% in HS rats and 42–69% in humans. Thus, the genetic architecture is broadly similar between rats and humans; the slightly lower cross-tissue correlations in humans (Fig 2D and 2E) could be due to a number of factors, including different collections of brain regions that were available for analysis in rats and humans.

## PrediXcan/TWAS extension to Rats (RatXcan) using mixed effects modeling

Having established the similarity between the genetic architecture of gene expression between rats and humans, we developed the RatXcan framework, extending the PrediXcan/TWAS framework from humans to rats. The software implementation is publicly available on a GitHub repository (https://github.com/hakyimlab/rat_genomics_paper_pipeline_2024). We used the predicted weights from the *training stage* to estimate the genetically regulated expression in the *target set* of 5,401 densely genotyped HS rats. We then tested the association between predicted expression and body length and BMI in the target set of rats.

Due to the high relatedness of the HS rats, simple linear regression approaches yield highly inflated association statistics. We confirmed that this was the case by performing a simulation

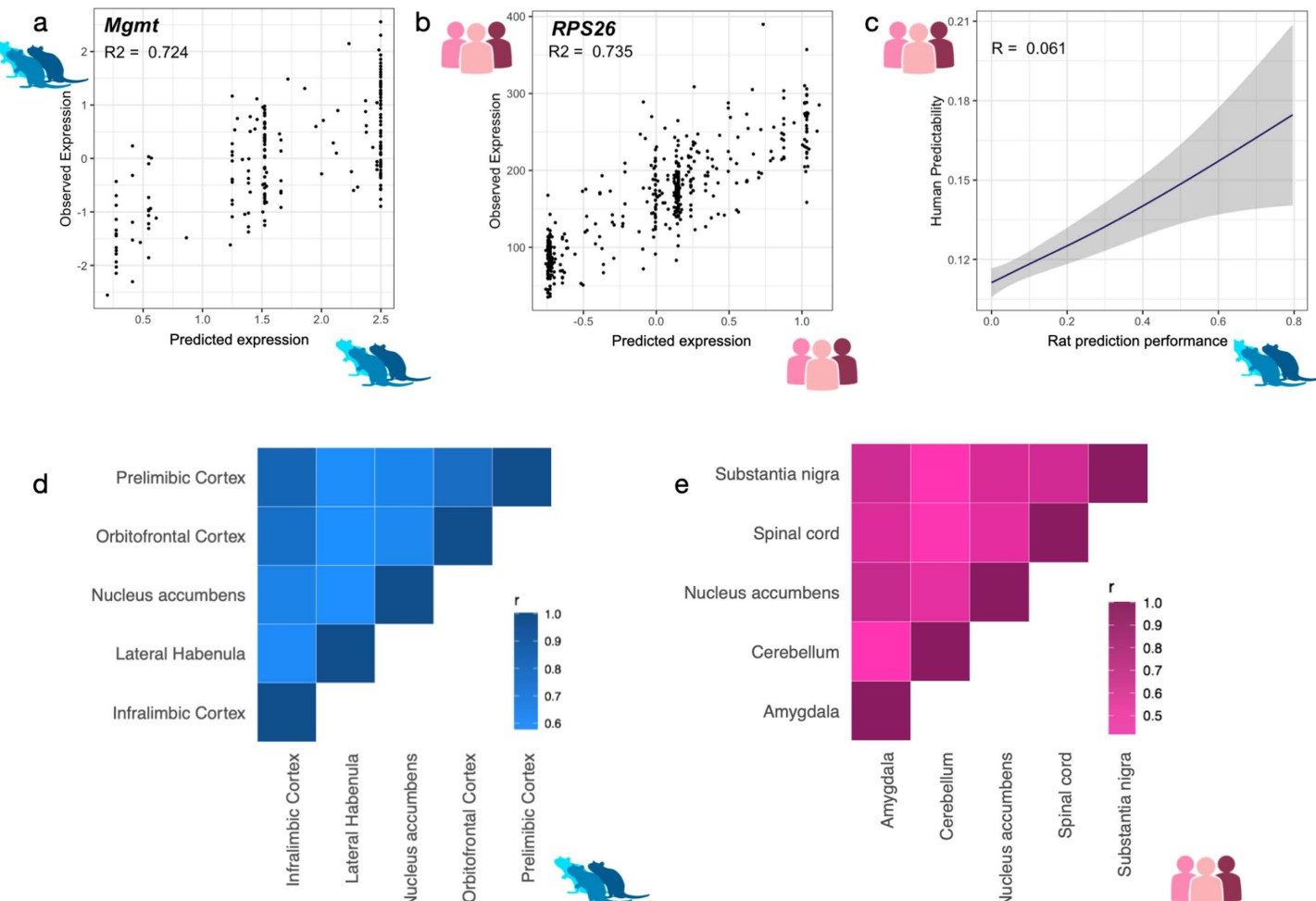

**Fig 2. Shared genetic architecture of gene expression in rats and humans a)** Comparison of predicted vs. observed expression for a well predicted gene in rats *Mgmt*, $R^2$ = 0.72, $R$ = 0.85, $P$ = 5.98 x $10^{-25}$). **b)** In humans, predicted and observed expression for *RPS26* were significantly correlated ($R^2$ = 0.74, $R$ = 0.86, $P$ = 2.13 x $10^{-30}$). **c)** Prediction performance (Pearson correlation) was significantly correlated across species ($R$ = 0.06, $P$ = 8.03 x $10^{-6}$). S3 Fig shows the corresponding scatter plot. **d-e)** and across all five brain tissues tested in rats and humans. In rats, within tissue prediction performance ranged from $R$ = 0.58 - 0.84 ($P$ = 9.85 x $10^{-20}$). In humans, the range was $R$ = 0.42 - 0.69 ($P$ = 8.25 x $10^{-19}$).

in which the effect of gene expression on the simulated phenotype $Y$ was 0, and we observed P-values that were concentrated below 0.05 as shown in Fig 3A. To account for the relatedness among individuals, we developed a mixed effects modeling approach where the genetic relatedness is represented as a random effect $u$, which has covariance that is equal to the genetic relatedness matrix ( $GRM$ ).

Although several mixed effect modeling approaches exist (e.g., GEMMA [20], GCTA [22], QTLRel [23], EMMAX [24], BoltLMM [25], they were designed for GWAS data. Here we developed a computationally efficient mixed effects method to associate genetically predicted expression with traits that accounts for relatedness.

We modeled the phenotype $Y$ as the sum of the contribution of gene expression $T$, weighted by effect size $b$, an individual-specific random effect $u$, and the usual independent noise term $\in$:

$$Y = T\,b + u + \in \tag{1}$$

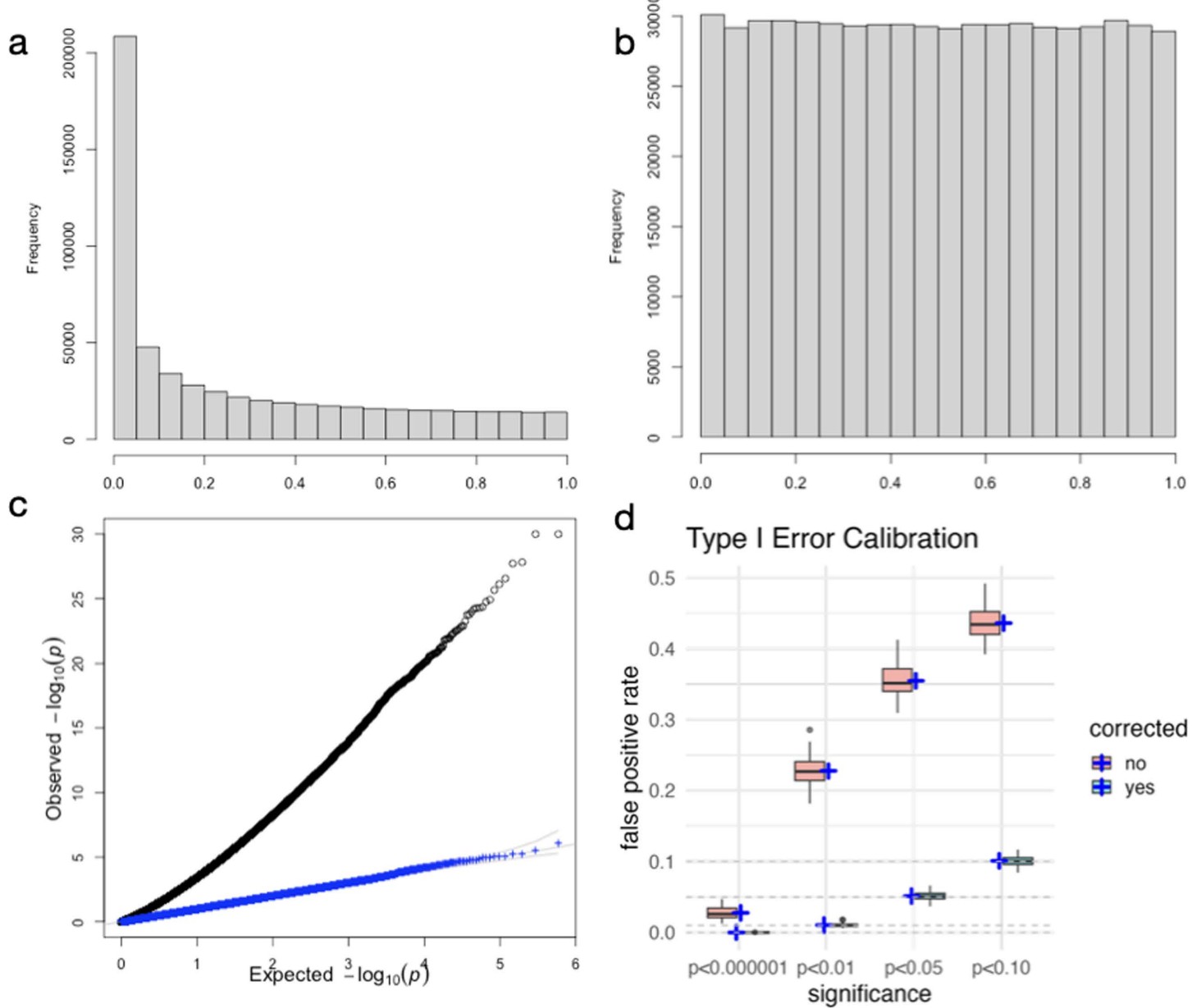

**Fig 3. Type I error calibration of RatXcan association results with relatedness correction.** a) Skewed distribution of P-values when phenotypes are simulated under the null ($Y = u + \epsilon$) and the gene expression to phenotype association is performed without accounting for relatedness. b) Uniform distribution of P-values with the mixed effects modeling approach, which corrects the inflation seen in a). c) QQ-plot of the p-values with (blue) and without (black) mixed effects correction. Blue dots follow the gray identity line, as expected under the null. **d)** Proportion of genes under the null (no relationship between phenotype and gene expression) with association P-value below $10^{-6}$, 0.01, 0.05, and 0.10. Reassuringly, for the green box plots corresponding to corrected P-values the proportion of tests below the stated threshold is centered around the threshold, i.e., ~1 in a million of tests yielded P-value < $10^{-6}$; ~1% of genes yielded P-value<0.01, ~5% of genes yielded P-values<0.05, ~10% of genes yielded a P-value<0.10. The pink box plots show uncorrected P-values with clear inflation. We used h2=0.40 for this Fig. The results are consistent across all heritability values in the range we tested (0.10 –0.80). Blue cross shows the average FPR (false positive rates). They fall right on the significance level for the corrected results.

where $\Sigma = \sigma^2(h^2\ GRM + (1 - h^2)I) = \sigma^2\Gamma$ is the covariance matrix of $u+\in$, where $GRM$ is the genetic relatedness matrix (based on genome-wide SNPs) and $I$ is the identity matrix, $\sigma^2$ is the variance of $Y$ under the null, $\sigma^2 h^2$ is the variance explained by the $GRM$, and

$\Gamma = h^2\ GRM + \left(1 - h^2\right)\boldsymbol{I}$. We linearly transformed the phenotype $Y$ and predicted expression $T$ by multiplying with the (matrix) square root of the unscaled covariance of the correlated error terms, $\Gamma^{-1/2}$ thereby decorrelating the error terms, which allowed us to simply use the traditional linear regression (see further details in Methods).

To demonstrate that our approach yields calibrated type I error, we performed a simulation study in which we simulated null phenotypes, $Y$, for 5,401 rats using $Y = u + \epsilon$. The random effect term $u$ can be simulated by multiplying the normal random variables vector with the (matrix) square root of the GRM; the independent noise term is simulated with normal random variables. We note that because of the deep connection between a random effect with covariance given by the GRM and a fully polygenic model [24], the correlated random effect, $u$, can also be represented or simulated as the sum of genotype dosages weighted by normally distributed random variables $\delta_k$,

$$\mu = \sum_k X_k \delta_k.$$

Because of this equivalence, which can be demonstrated by showing that the covariance matrices of both sides of the equality are the same, RatXcan accounts for relatedness and also corrects the polygenicity-driven inflation reported in [16]. The proportion of variance in the phenotype explained by the genotype—estimated as the SNP heritability—will include random effects that account for relatedness as well as a fully polygenic component $\sum_k X_k \delta_k$.

For each heritability value ranging from 0.1 to 0.8, we simulated 100 null phenotypes and performed the traditional and mixed effects model association between predicted expression and $Y$. Reassuringly, P-values from our mixed effects modeling approach are uniformly distributed (Fig 3B and 3C). Fig 3D shows that RatXcan yields a proportion of false positives matching the significance levels of 0.1, 0.05, 0.01, and $10^{-6}$, as expected. To get a better estimate of the proportion of false positives when using 1e-6 as a threshold, we combined 2.9 million tests (5,879 genes by 100 simulations by 5 heritability values) and found 3 tests below the threshold, yielding a false positive rate of $1.02 \times 10^{-6}$ and further confirming the calibration of our corrected test. Simulating $u$ as a correlated random variable or as a weighted sum of genotype dosages with normally distributed effect sizes yielded the same calibration results.

We also investigated the effect of pruning for LD before calculating GRM or removing variants in the proximity of the tested gene and found that they do not completely correct the inflation (see S4 and S5 Figs). Hence, we recommend using genome-wide SNPs without pruning for LD or filtering out proximal SNPs.

After verifying that our RatXcan associations had calibrated type I error (as shown in Fig 3B where p-values are uniformly distributed under the null), we applied this methodology to body length and BMI phenotypes in rats. To increase the coverage of predicted genes, we combined predicted expression across all five tissues using the ACAT method [26], yielding 10,770 genes tested in at least one of the prediction models. We used ACAT because of its robustness to misspecified correlations.

We identified 11 Bonferroni significant genes ($P\ (0.05/8272)=6.04 \times 10^{-6}$) in three loci on chromosomes 7, 10, and 14 for rat body length (Fig 4A) and 10 significant genes in three loci on chromosomes 1, 10, and 18 for rat BMI (Fig 4C and S1 Table). Among the top significant genes, prolactin-releasing hormone receptor *Prlhr* was associated with BMI (P=1.52 x 10⁻⁹). *Prlhr* has been previously implicated in obesity and energy expenditure in mice [27,28] and was associated with BMI in a prior GWAS using a separate cohort of HS rats [9,29] and in a GWAS that used a subset of the rats used in the current study [8]. The human ortholog, *PRLHR*, was also associated with BMI (P = 1.76 x 10⁻⁶) and body fat percentage (P = 3.62 x

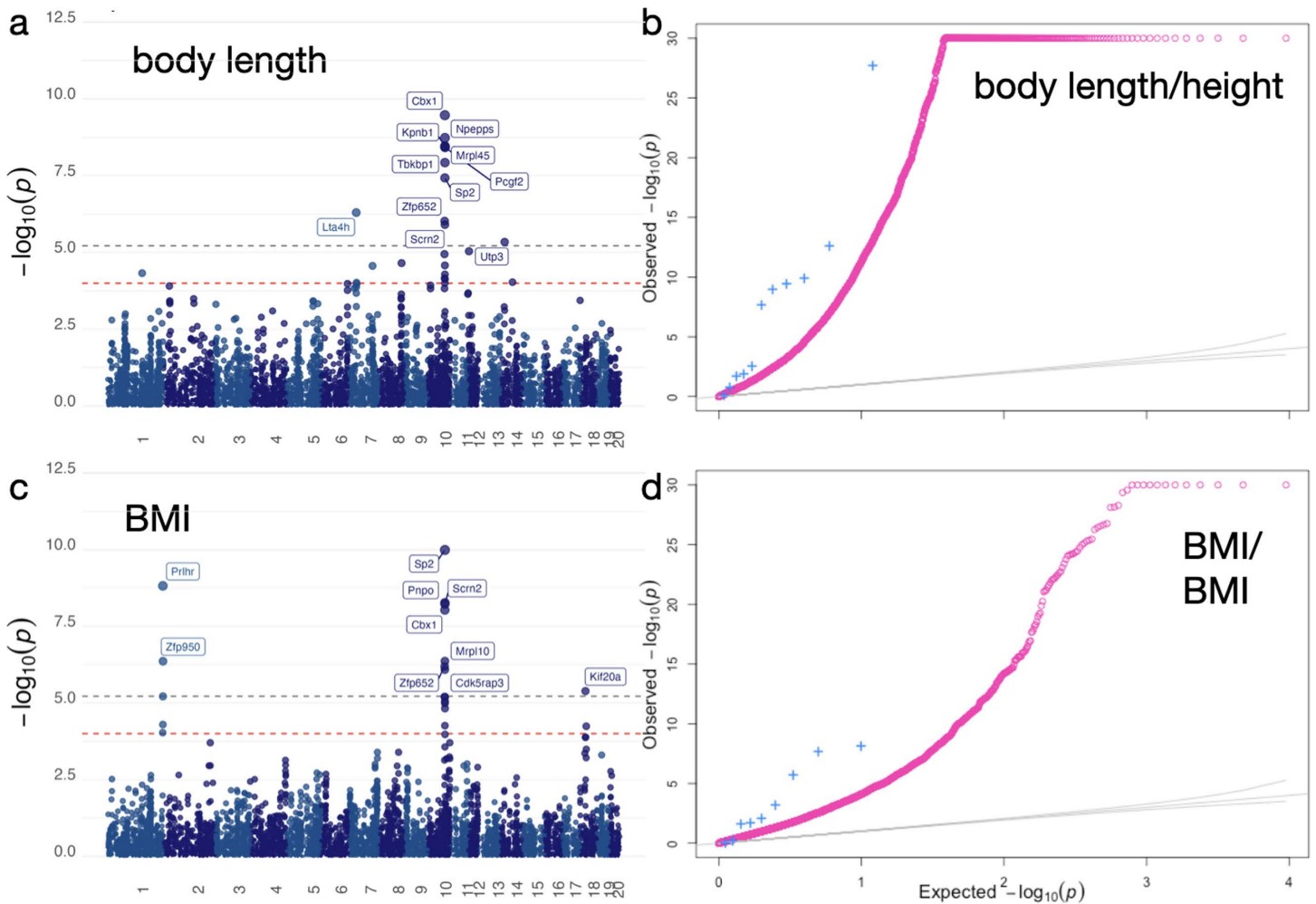

**Fig 4. RatXcan association and enrichment.** a) Manhattan plot of the association between predicted gene expression and rat body length, which is analogous to human height. Association results with the 5 tissues combined into one p-value using the ACAT approach. b) Q-Q plot of the *P*-values of the association between predicted gene expression levels and height in humans (phenomexcan.org). Pink dots correspond to all genes tested. Blue crosses correspond to the subset of genes that were significantly associated with body length in rats (Fisher test, *P*=0.016). c) Manhattan plot of the association between predicted gene expression and rat BMI. In both a) and c) we label Bonferroni significant genes. Gray dotted line corresponds to the Bonferroni correction threshold of 0.05/5,388 of tests. Red dotted line corresponds to a threshold of 1 x 10⁻⁴. d) Q-Q plot of the *P*-values of the association between predicted gene expression levels and BMI in humans (phenomexcan.org). Pink dots correspond to all genes. Blue crosses correspond to the subset of genes that were significantly associated with BMI in rats (Fisher test, *P*=0.013).

$10^{-6}$) [19] in TWAS in humans. Our *Prlhr* association with BMI adds to the multiple lines of evidence and further reinforces the promise of *PRLHR* as target for obesity treatment [28]. The complete list of results for body length and BMI are listed in S1 and S2 Tables and are also available at http://imlab.shinyapps.io/RatXcan.

To evaluate whether trait-associated genes in rats were significantly associated with the corresponding trait in humans, we performed enrichment analysis. Specifically, we selected genes that were significantly associated with rat body length ($P <$ 0.05/number of tests) and compared the *P*-values from the analogous human trait (height) against the background distribution. The background distribution (pink, Fig 4B) of *P*-values for the association between rat body length genes and human height depart substantially from the identity line (gray), which is expected given the large sample size of the human height GWAS. The subset of genes that were associated with rat body length (blue, Fig 4B) showed a departure from the background

distribution (Fisher test, $P$=0.016), indicating that body length genes in rats were significantly enriched among human height genes. We repeated the analysis for rat BMI genes and likewise found enrichment in human BMI (Fisher test, $P$=0.013).

## Methods

### Ethics statement

Our research using de-identified human data which has been determined to be non-human subject by the University of Chicago IRB under protocol IRB16–0980. Regarding the rat data, we perform secondary analysis of publicly available data. The original data generators have obtained approval of the procedures from their respective institutional animal care and use committee (IACUC).

### Experimental model and subject details

The rats used for this study are part of a large multi-site project focused on genetic analysis of complex traits (www.ratgenes.org). Outbred HS rats are the most highly recombinant rat intercross available and are a powerful tool for genetic studies [15]. HS rats were created in 1984 by interbreeding eight inbred rat strains (ACI/N, BN/SsN, BUF/N, F344/N, M520/N, MR/N, WKY/N and WN/N) and been maintained as an outbred population for 100 generations. The rat BMI (weight/height^2) and body length (including tail) data used in this analysis consist of the individuals used in a prior study of 3,173 [8] as well as 2,228 additional HS rats, many of which were also used in [18]. These 5,401 rats were produced by a breeding colony at the Medical College of Wisconsin (NMcwi:HS #2314009, RRID:RGD_2314009) and had been subjected to various behavioral treatments, as described in [8]. For each trait, sex, age, batch number, and site were regressed out if they were significant and if they explained more than 2% of the variance, as described in [8].

### Genotype and expression data in the training rat set

For training the gene expression predictors (Table 1), we used RNAseq and genotype data from 88 HS rats that were pre-processed by [17]. The mean age of these HS rats was 85.7 ± 2.2 days for males and 87.0 ± 3.8 for females. Prior to tissue collection, these 88 rats were group housed under standard laboratory conditions and had not been subjected to any previous treatments or experimental protocols. Genotypes were determined using genotyping-by-sequencing, as described previously [8,17,30]. Bulk RNA-sequencing was performed using Illumina HiSeq 4000 with polyA libraries, 100 bp single-end reads, and mean library size of ~27M. Read alignment and gene expression quantification were performed using RSEM and counts were upper-quartile normalized, followed by additional quality-control filtering steps as described in [17]. Gene-expression levels refer to transcript abundance for reads aligned to the gene's exons using the Ensembl Rat Transcriptome release 99 (Rnor_6.0).

For each gene, we inverse normalized the TPM (transcripts per million) values to minimize the effects of outliers and fit a normal distribution. We filtered out genes that did not pass the Shapiro test for normality since after inverse normalization continuous random variables must follow normal distribution exactly (failure can be due to excessive ties, indication of many 0's). We then computed the principal components to estimate unwanted variation [31]. We regressed out sex, batch number, and the 7 top gene expression principal components and saved the residuals for all downstream analyses.

### Querying human gene-trait association results

To retrieve analogous human gene-trait association results, we queried PhenomeXcan, a web-based tool that provides gene-level association results for 4,091 traits based on predicted expression in 49 GTEx tissues [19]. Orthologous genes (N = 22,777) were mapped with Ensembl annotation, using the *biomart R* package.

```
orth.rats = getBM(attributes = c("ensembl_gene_id", "external_
gene_name", "rnorvegicus_homolog_ensembl_gene","rnorvegicus_
homolog_associated_gene_name"),filters="with_rnorvegicus_
homolog",values=TRUE, mart = human, uniqueRows=TRUE)
```

### Estimating gene expression heritability

We calculated the cis-heritability of gene expression from the training set using a Bayesian sparse linear mixed model, BSLMM [20], as implemented in GEMMA. We used variants within the ±1Mb window up- and downstream of the transcription start and end of each gene annotated by Ensembl release 99 rat annotations. We used the proportion of variance explained (**PVE**) generated by GEMMA as the measure of cis-heritability of gene expression. We then displayed only the PVE estimates of 10,268 genes that were also present in the human gene expression data.

Heritability of human gene expression, which was also calculated with GEMMA, was downloaded from the database generated by [21]. Genes were limited to the same 10,268 as above.

### Examining polygenicity versus sparsity of gene expression

To examine the polygenicity versus sparsity of gene expression in HS rats, we identified the optimal elastic net mixing parameter α, as described in [21]. Briefly, we compared the prediction performance of a range of elastic net mixing parameters spanning from 0 to 1 (11 values from 0 to 1, with steps of 0.1). If the optimal mixing parameter was closer to 0, corresponding to ridge regression, we deemed the gene expression trait to be polygenic. In contrast, if the optimal mixing parameter was closer to 1, corresponding to lasso, then the gene expression trait was considered to be more sparse. We restricted the number of genes in the pipeline to the 10,268 orthologous genes using *biomart R*, as described above.

### Training gene expression prediction in rats

To train prediction models for gene expression in HS rats, we used the training set of HS rats from [17] and followed the elastic net pipeline from predictdb.org. Briefly, for each gene, we fitted an elastic net regression using the *glmnet* package in R. We only included variants in the cis region (i.e., 1Mb up and downstream of the transcription start and end). The regression coefficient from the best penalty parameter (chosen via *glmnet*'s internal 10-fold cross validation [32] served as the weight for each gene. The calculated weights ($w_s$) are available in predictdb.org.

### Estimating overlap and enrichment of genes between rats and humans

For human transcriptome prediction used in the comparison with rats, we downloaded elastic net predictors trained in GTEx whole blood samples from the PredictDB portal [33]. Using brain predictors yielded similar results.

We quantified the accuracy of the prediction models using a 10-fold cross-validated correlation ($R$) and correlation squared ($R^2$) between predicted and observed gene expression [32]. For the rat prediction models, we only included genes whose prediction performance

was greater than 0.01 and had a non-negative correlation coefficient, as these genes were considered well predicted.

We tested the prediction performance of our elastic net model trained in NAcc in an independent rat reference transcriptome set of 188 NAcc samples that was downloaded from RatGTEx.

## RatXcan framework

We developed RatXcan, extending PrediXcan [6, 16, 34] to predict the association between rat gene expression and human traits. For prediction of rat gene expression, RatXcan uses the elastic net prediction models generated in the training set. In the association stage, we computed the genetically predicted expression matrix for all genes in the rat target set, as a linear combination of genotype dosages $X_k$ and the weights from the training stage $\omega_{kg}$

$$T_g = \sum_k \omega_{k,g} X_k$$

We then tested the association between the predicted expression matrix and the traits (body length and BMI). To account for the relatedness across individuals, we fitted a mixed effects model

$$Y = T\ b + u + \in \tag{2}$$

where $Y$ is the trait, $T$ is the expression level of a gene (the subscript g is dropped here), $b$ is the effect of the gene to be estimated, $u$ is the random effect with covariance given by the $GRM$, representing the correlation across rats due to the relatedness, and $\in$ is the usual uncorrelated noise.

Fitting mixed effects models can be computationally expensive. To make estimation more computationally efficient, we transformed the phenotype and the predicted expression such that the resulting noise term becomes uncorrelated. To achieve this goal, we used the following approach.

We decorrelated the error term $u + \in$ by premultiplying $Y$ with where $\Gamma^{-1/2} = (\Sigma/\sigma^2)^{-1/2}$ is the covariance matrix of $u + \in$, i.e.,

$$\Sigma = \sigma^2(h^2\ GRM + \left(1 - h^2\right)\boldsymbol{I})$$

where GRM is the genetic relatedness matrix and I is the identity matrix, $\sigma^2 h^2$ is the variance explained by the $GRM$ (estimable with GCTA as the heritability or similar software), $\sigma^2$ is the variance of $Y$ under the null. We defined the GRM as in the GCTA paper [22] such that the genetic relatedness between individual $i$ and $j$ is given by

$$GRM_{ij} = \frac{1}{M}\sum_k^M \frac{\left(X_{ik} - 2p_k\right)\left(X_{jk} - 2p_k\right)}{2p_k\left(1 - p_k\right)}$$

$M$ is the total number of SNPs considered, $p_k$ the population allele frequency of SNP $k$.

The transformed phenotype $\Gamma^{-1/2} \cdot Y$ has uncorrelated error terms by multiplying both sides of (eq 2) with $\Gamma^{-1/2}$

$$\Gamma^{-1/2} \bullet Y = \Gamma^{-1/2}\ T \bullet b + \Gamma^{-1/2} \bullet \left(u + \in\right) \tag{3}$$

The transformed noise term $\Gamma^{-1/2}.(u+\in)$ has a covariance, which is proportional to the identity matrix as shown next. The covariance of $\Gamma^{-1/2}.(u+\in)$ is given by

$$E\left[\left(\Gamma^{-1/2}.(u+\in)\right).\left(\Gamma^{-1/2}.(u+\in)\right)'\right]=$$

$$E\left[\Gamma^{-1/2}.(u+\in).(u+\in)'.\Gamma^{-1/2}\right]=$$

$$\Gamma^{-1/2}.E\left[(u+\in).(u+\in)'\right].\Gamma^{-1/2}=, \text{ using } (A.B)'=B'.A' \text{ and } \Gamma=\Gamma' z$$

$\Gamma^{-1/2}.\sigma^2\Gamma.\Gamma^{-1/2}=\sigma^2 I$ , using that the covariance matrix of $u+\epsilon$
is $E\left[(\mu+\in).(\mu+\in)'\right]=\sum=\sigma^2\Gamma$

We can rewrite equation 3 in terms of the transformed variables
$\tilde{Y}=\Gamma^{-1/2}.Y, \tilde{T}=\Gamma^{-1/2}.T \text{ and } \tilde{\varepsilon}=\Gamma^{-1/2}.(\mu+\in)$, we get

$$\tilde{Y}=\tilde{T}b+\tilde{\varepsilon} \tag{4}$$

In the transformed space, the errors become uncorrelated, and therefore, we can estimate the effect size $b$ using the regular linear regression approach.

### Estimating overlap and enrichment of genes between rats and humans

We queried PhenomeXcan to identify genes associated with human height and BMI. PhenomeXcan provides gene-level associations aggregated across all available GTEx tissues, as calculated by MultiXcan (an extension of PrediXcan) [35]. For the rat gene associations, we aggregated our results across the five tested brain regions using the ACAT method, which is a more robust approach than MultiXcan because it does not depend on correlation estimations that can be misspecified. We used a Q-Q plot to inspect the level of enrichment across rat and human findings. To quantify enrichment, we used a Fisher test to assess whether rat trait-associated genes were also likely to be associated in humans.

### Discussion

We present RatXcan, which is an extension of PrediXcan, a well-established statistical framework that is used in human genetics to link genes to phenotypes [6,16], that connects predicted rat gene expression to human traits associated with orthologous genes. RatXcan is a computationally efficient method that corrects for the inflation due to polygenicity and relatedness of the rats [16] using a mixed effects approach. We showed that the genetic architecture of gene expression in rats is broadly similar to humans: they are both heritable and sparse, and the degree of heritability is preserved across tissues; some of these observations are consistent with another recent publication that mapped eQTLs in HS rats [17].

We found higher heritability estimates for gene expression traits in rats compared to typical human studies, which could be due to the fact that our rat cohorts are likely to be subjected to a more homogeneous environment than an equivalently sized human cohort, which will lead to higher heritability (i.e., smaller denominator driven by heritable and environmental factors). Another factor likely increasing our estimates of rat heritability is the relatedness of the HS rats, despite our effort to select distantly related ones.

Using RatXcan, we tested gene-level associations of body length/height and BMI, which had been previously measured in rats. We chose height and BMI because of the availability of large human GWAS, a relatively large genotyped HS rat cohort in which body length and weight were known, and relatively unambiguous similarity between the human and rat traits. We found significant enrichment of trait-associated genes among orthologous human trait-associated genes. Our data provided urgently needed empirical data supporting the genetic similarity of traits in rodents and humans that helps address the ongoing debate about the validity of genetic animal models of human traits. While our approach is very different, we reached a similar conclusion in another recent publication that also explored polygenic similarities between HS rats and humans [18].

This mixed effects modeling approach implemented in RatXcan can be applied to human and other species TWAS when individual-level data are available. However, for biobank-scale data, we recommend using the summary statistics-based method in [16]. S-PrediXcan and other summary statistics-based methods that do not address the inflation driven by polygenicity and relatedness as described here and in [16] will yield higher false positive rates than expected.

Overwhelming evidence demonstrates that most complex diseases are extremely polygenic; however, translating these findings into biologically meaningful discoveries is challenging. Furthermore, there is an unmet need for methods that translate polygenic results between species. The data produced by human GWAS provide information about the role of individual SNPs in conveying risk; however, SNPs do not have direct homologs across species, and even if they did, they would not be expected to have the same effects or to tag the same causal variants. For these reasons, GWAS results are not amenable to cross-species integration. Instead, efforts at cross-species translation have focused on using non-human organisms to study the role of *individual* genes [36]. Although valuable, these approaches are unable to capture the *polygenic* liability identified in human GWAS. Furthermore, the alleles studied in model systems are typically loss-of-function alleles, which may be qualitatively different from the relatively subtle, small effect variants typically identified in human GWAS. The inability to model polygenic vulnerability using animals is a major impediment to progress and has been a topic of active discussion [13]. RatXcan addresses these issues by simultaneously circumventing the limitation of using SNPs and encompassing the polygenicity found in most complex traits by holistically mapping orthologous genes between the model species and humans.

There are several limitations in the current study. The sample size of the reference transcriptome data in rats was limited. We would expect better prediction performance in our elastic-net trained models with larger sample sizes. Furthermore, we used gene expression data from human blood and rat nucleus accumbens core because they were convenient datasets, but these tissues are not necessarily the most appropriate for traits like height or BMI. Second, we suspect that in both rats and humans, some gene-level associations may be confounded by linkage disequilibrium contamination and co-regulation. This problem is likely to be more serious in model organisms where LD is more extensive. Third, our method depends on having access to individual-level data and needs to compute the GRM and the eigenvector decomposition, which may limit the application to medium sample sizes (under ~50K). For larger sample sizes, or for samples for which individual genotype data are not available, we recommend using the method in [16], which can be applied using GWAS summary statistics. Finally, integration of other omic data types (e.g., protein, methylation, metabolomics) and the use of cell-specific data may improve cross-species portability. It is worth noting that while we have shown success with humans and HS rats, it is still not clear whether more distantly related species, such as non-mammalian vertebrates or even insects, might also lend themselves to a similar analysis.

Despite these limitations, we have developed a methodology for effectively and efficiently identifying overlapping polygenic architecture between rats and humans. Our results provide a method to empirically validate traits that are intended to model or recapitulate aspects of human diseases in model systems and support experimental designs whereby genetic information from model organisms and humans can be better integrated, thus enabling a greater biological understanding of human GWAS results and, by extension, human disease.

## Supporting information

**S1 Fig. Gene expression was heritable 8.86-10.12% and comparable across several brain tissues tested (Infralimbic Cortex, IL; Lateral Habenula, LHb; Prelimibic Cortex, PL; Orbitofrontal Cortex, OFC) in rats.** We refer to heritability (h2, cis-heritability within 1Mb) as the proportion of variance explained (PVE). Across all brain tissues tested, heritability estimates were significantly correlated ($R$ = 0.58-0.83, $P$ = 3.14 x $10^{-19}$).
(TIFF)

**S2 Fig. Heritability of gene expression was correlated between rats and humans.** We found a significant correlation (R = 0.07, P = 4.34 x 10-12) between heritability estimates in rats and humans. Confidence intervals are represented as gray bars. The gray line represents the null distribution. Top panel shows the smoothed lines with loess (Local Polynomial Regression Fitting) implemented in the ggplot2 package in R. The bottom panel shows the same Fig with all the points in addition to the smoothed curve.
(TIFF)

**S3 Fig. Shared genetic architecture of gene expression in rats and humans Prediction performance in humans vs rats.** The performance measure (Pearson correlation) was significantly correlated across species (R = 0.06, P = 8.03 x 10–6).
(TIFF)

**S4 Fig. Leaving one chromosome out for GRM calculation under corrects the inflation.** Using the same null trait simulation used in Fig 3, we performed RatXcan association for genes in chromosome 1 using a GRM calculated excluding variants in chromosome 1. Results for genes in chromosome 1 are shown. A well corrected QQ-plot should have all points on the gray diagonal line but we observed apparent inflation. Hence LOCO for GRM calculation is not recommended.
(TIFF)

**S5 Fig. Calculating GRM with LD pruned variants reduces the effectiveness of the correction.** Using the same null trait simulation shown in Fig 3, we performed RatXcan using an LD-pruned GRM. Pruning was done using plink with –indep-pairwise 500 5 0.95, which retains variants with r2 smaller than 0.95 using window size of 500Kb and shifting the window 5 variants at a time. A well corrected QQ-plot should have all points on the gray diagonal line but we observed apparent inflation. Hence LD-pruning is not recommended.
(TIFF)

**S1 Table. Body length association with predicted gene expression for the 5 brain regions (Infralimbic Cortex, IL; Lateral Habenula, LHb; Prelimibic Cortex, PL; Orbitofrontal Cortex, OFC).** Column name annotation, gene_name: gene name, p_acat_5: Combined p-values across 5 brain regions using the ACAT method, chr: chromosome, start: start position of the gene, qval: qvalue calculated with the qvalue package, p_human: p-value of the association in humans of the mapped human trait (phenomexcan.org), hugo_gene: mapped human gene name, trait: rat trait name, gene: rat gene ensembl id, gene_id: mapped human gene ensembl

id, AC: p-value of association with predicted expression in Nucleus Accumbens, IL: p-value of association with predicted expression in Infralimbic Cortex, LH: p-value of association with predicted expression in Lateral Habenula, PL: p-value of association with predicted expression in Prelimbic Cortex, OFC: p-value of association with predicted expression in Orbitofrontal Cortex.
(XLSX)

**S2 Table. Body Mass Index association with predicted gene expression for the 5 brain regions (Infralimbic Cortex, IL; Lateral Habenula, LHb; Prelimbic Cortex, PL; Orbitofrontal Cortex, OFC). Column name annotation is the same as S1 Table.**
(XLSX)

## Acknowledgements

This research used resources of the Argonne Leadership Computing Facility, which is a DOE Office of Science User Facility supported under Contract DE-AC02–06CH11357. This work was completed in part with resources provided by the University of Chicago's Research Computing Center and Beagle3. We also acknowledge resources from the Center for Research Informatics, funded by the Biological Sciences Division at the University of Chicago, with additional funding provided by the Institute for Translational Medicine and the CTSA grant number 2U54TR002389–06.

## Author contributions

**Conceptualization:** Sandra Sanchez-Roige, Abraham A. Palmer, Hae Kyung Im.

**Data curation:** Apurva Chitre, Daniel Munro, Denghui Chen, Angel Garcia-Martinez, Anthony M. George, Alexander F Gileta, Wenyan Han, Katie Holl, Alesa Hughson, Christopher P. King, Alexander C. Lamparelli, Connor D. Martin, Festus Nyasimi, Celine L. St. Pierre, Sarah Sumner, Jordan Tripi, Tengfei Wang.

**Formal analysis:** Natasha Santhanam, Sandra Sanchez-Roige, Sabrina Mi, Yanyu Liang, Festus Nyasimi, Hae Kyung Im.

**Funding acquisition:** Hao Chen, Shelly Flagel, Paul Meyer, Oksana Polesskaya, Laura Saba, Leah C. Solberg Woods, Abraham A. Palmer, Hae Kyung Im.

**Investigation:** Apurva Chitre, Daniel Munro, Denghui Chen, Angel Garcia-Martinez, Anthony M. George, Alexander F Gileta, Wenyan Han, Katie Holl, Alesa Hughson, Christopher P. King, Alexander C. Lamparelli, Connor D. Martin, Festus Nyasimi, Celine L. St. Pierre, Sarah Sumner, Jordan Tripi, Tengfei Wang.

**Methodology:** Natasha Santhanam, Sandra Sanchez-Roige, Abraham A. Palmer, Hae Kyung Im.

**Project administration:** Sandra Sanchez-Roige, Hao Chen, Shelly Flagel, Paul Meyer, Oksana Polesskaya, Laura Saba, Leah C. Solberg Woods, Abraham A. Palmer, Hae Kyung Im.

**Resources:** Abraham A. Palmer, Hae Kyung Im.

**Software:** Natasha Santhanam, Sandra Sanchez-Roige.

**Supervision:** Sandra Sanchez-Roige, Shelly Flagel, Keita Ishiwari, Paul Meyer, Oksana Polesskaya, Laura Saba, Leah C. Solberg Woods, Abraham A. Palmer, Hae Kyung Im.

**Validation:** Natasha Santhanam, Sandra Sanchez-Roige.

**Visualization:** Natasha Santhanam, Sabrina Mi.

**Writing – original draft:** Natasha Santhanam, Sandra Sanchez-Roige, Abraham A. Palmer, Hae Kyung Im.

**Writing – review & editing:** Natasha Santhanam, Sandra Sanchez-Roige, Sabrina Mi, Yanyu Liang, Apurva Chitre, Daniel Munro, Denghui Chen, Jianjun Gao, Angel Garcia-Martinez, Anthony M. George, Alexander F Gileta, Wenyan Han, Katie Holl, Alesa Hughson, Christopher P. King, Alexander C. Lamparelli, Connor D. Martin, Festus Nyasimi, Celine L. St. Pierre, Sarah Sumner, Jordan Tripi, Tengfei Wang, Hao Chen, Shelly Flagel, Keita Ishiwari, Paul Meyer, Oksana Polesskaya, Laura Saba, Leah C. Solberg Woods, Abraham A. Palmer, Hae Kyung Im.

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
