## [Decision Letter · Decision Letter 0]

9 Aug 2024

Dear Dr Im,

Thank you very much for submitting your Research Article entitled 'RatXcan: A framework for cross-species integration of genome-wide association and gene expression data' to PLOS Genetics.

The manuscript was fully evaluated at the editorial level and by independent peer reviewers. The reviewers appreciated the attention to an important problem, but raised some substantial concerns about the current manuscript. Based on the reviews, we will not be able to accept this version of the manuscript, but we would be willing to review a much-revised version. We cannot, of course, promise publication at that time.

If you decide to revise the manuscript for further consideration at PLOS Genetics, please aim to resubmit within the next 60 days, unless it will take extra time to address the concerns of the reviewers, in which case we would appreciate an expected resubmission date by email to plosgenetics@plos.org.

If present, accompanying reviewer attachments are included with this email; please notify the journal office if any appear to be missing. They will also be available for download from the link below. You can use this link to log into the system when you are ready to submit a revised version, having first consulted our Submission Checklist .

PLOS has incorporated Similarity Check , powered by iThenticate, into its journal-wide submission system in order to screen submitted content for originality before publication. Each PLOS journal undertakes screening on a proportion of submitted articles. You will be contacted if needed following the screening process.

To resubmit, log into your Editorial Manager account and select the option 'Revise Submission' in the 'Submissions Needing Revision' folder.

We are sorry that we cannot be more positive about your manuscript at this stage. Please do not hesitate to contact us if you have any concerns or questions.

Yours sincerely,

Jingjing Yang, Ph.D.

Academic Editor

PLOS Genetics

Xiaofeng Zhu

Section Editor

PLOS Genetics

All reviewers are quite positive and find the authors' work is important for the field. Please address all reviewers' comments in your major revision.

Reviewer's Responses to Questions

**Comments to the Authors:**

Reviewer #1: Santhanam et al. present RatXcan, a framework for PrediXcan-like analysis with samples of related individuals, demonstrating its utility for cross-species integration of genome-wide association and gene expression data. The manuscript is engaging and well-structured. While the study focuses on model organisms, as an animal quantitative geneticist, I believe RatXcan has significant potential for GTEx studies in farm animals. I have several minor comments and questions that may help broaden the study's impact:

1. Heritability and prediction models: The study uses variants within ±1Mb windows up- and downstream of each gene's transcription start and end sites for heritability estimation and prediction modeling, without accounting for relatedness using a GRM. The manuscript has well justified the use of GRM in RatXcan associations. I wonder if modeling relatedness is also necessary for prediction and cis-heritability estimation. In such cases, could a GRM be constructed excluding the variants within the ±1Mb window?

2. GRM construction for RatXcan associations: The manuscript does not explicitly state which variants are used in the GRM for RatXcan associations. (Apologies if I overlooked this information.) Are all available variants used, or are those within the ±1Mb window for each focal gene excluded? Alternatively, are LD-pruned variants employed? Using whole-genome LD-pruned variants for GRM construction seems a straightforward approach that could control inflation, despite potential proximal contamination.

3. Applicability of RatXcan vs. PrediXcan and S-PrediXcan: It appears that RatXcan is suitable for samples of highly related individuals, while PrediXcan is not. Could you clarify the applicability of S-PrediXcan in this context? This information would be particularly valuable for farm animal genetics researchers. Several GTEx projects for farm animal species (e.g., PMIDs: 35953587, 38177344) have used S-PrediXcan, which seems potentially inappropriate given that S-PrediXcan, like PrediXcan, is designed for samples of unrelated individuals. Considering that individuals in cattle or pig populations are often highly related due to strong artificial selection and extensive use of artificial insemination (elite male animals may have tens of thousands of offspring), there's a concern about the potential misuse of S-PrediXcan in these farm animal GTEx projects. A clarification of this issue in the Discussion section would be greatly appreciated.

Reviewer #2: In this manuscript, the authors extended the PrediXcan methodology to outbred heterogeneous stock (HS) rats. They developed RatXcan to account for close familial relationships among HS rats, trained transcript predictors for about 9,000 genes using reference genotype and expression data from five rat brain regions, and tested the association between predicted expression and body length and BMI in 5,401 densely genotyped HS rats. The manuscript is well structured and clearly written. I have a few comments and suggestions for the authors, which I listed below.

1. I think eq 2b (page 17) was not really a standard linear regression model in the sense that the residual variance (\tilde\epsilon) was fixed as 1. Perhaps a better “decorrelation” approach would be to left multiply Y and T by (h^2GRM+(1-h^2)I)^(-1/2) which does not include \sigma^2, so that the residual variance in eq 2b can be estimated from a standard linear regression model.

2. Were 88 HS rats in the training set assumed to be completely unrelated? It was unclear whether and how relatedness was adjusted for in the elastic net, and some clarifications would be helpful.

3. In addition to 0.1, 0.05, 0.01, it would be great to evaluate the type I error rates of RatXcan at much lower significance levels, since the method was applied to real data at the Bonferroni-corrected significance level of 0.05/8,272=6.04e-6.

Minor:

1. Figure 2a legend showed R^2=0.72 and R=0.65. It could be a typo (R=0.85?). Also most p-values in Figure 2 legend were shown as P < 2.2e-16 which was not very informative. It would be nice to show lower p-values with a better precision.

2. Figure 2c was not very informative. It was unclear whether “prediction performance” was defined using R^2 or R. It would be nice to add the data points in a scatter plot, in addition to the line and confidence band. Similarly, it would be nice to show heritability data points in Figure S2.

3. In the second paragraph on page 11, (P=1.76e-6; P=3.62e-6) looked a little confusing. It would be better to place each p-value right after “BMI” and “body fat percentage” using two separate parentheses.

4. Methods – Experimental model and subject details: “a prior study of 3,173 as well as more than 2,000 additional HS rats”. Please give the exact number of “additional HS rats” if possible.

5. Methods – last paragraph on page 13: TPM should be defined.

Reviewer #3: In the manuscript under consideration, the authors presented RatXcan, an extension of their TWAS (PrediXcan) to outbred heterogeneous stock (HS) rats. The main innovation is to use a linear mixed model (eq1 on page 16 where a random effect term is added) to conduct the association mapping between predicted gene expressions and the phenotype in a typical TWAS protocol. By the addition of the random term, the infinitesimal polygenic effect is captured therefore the inflated p-value due to population structure (or uneven relatedness) is controlled. The LMM is solved by standard de-correlation procedure in which the data are transformed and then associated.

The works looks intuitively sound, and the mathematical derivations are all correct for me. The real data analysis looks routine and solid for me as well. I am happy to endorse its publication. I have the following suggestions for the authors to consider.

An important concern is on the problem of not using LMM in PrediXcan back to their 2015 Nat Genet paper. Given the popularity of LMM (Kang et al 2010 Nat Genet; and a few others) in GWAS before the publication of PrediXcan, it is counterintuitive that PrediXcan does not adapt LMM in the first place, causing the massive inflation of p-values. The present work acts as a nice remedy to this problem. This is an inherent problem of all TWAS analysis for human cohorts where population structure is not avoidable, instead of just in HS rats. I feel that the authors may be explicit on this issue and explain that the current work may be used to replace PrediXcan (instead of just for rats).

The average heritability of gene expressions (Table 1) is way higher than it is in humans. Any evidence showring this estimate is correct from other literatures? From the materials section, I see that only 88 rats’ data are available. Will this small sample size cause problem to an overestimate of expression heritability?

The work (Liang et al 2023) has been cited many times in the main text, however I can’t find it in the references section. This actually serves as critical evidence for the claims of resolving p-value inflations. Nevertheless, I tend to trust this work indeed provides the claimed evidence, as intuitively I agree with the authors’ argument on this point.

**Have all data underlying the figures and results presented in the manuscript been provided?**

Reviewer #1: Yes

Reviewer #2: Yes

Reviewer #3: Yes

PLOS authors have the option to publish the peer review history of their article (what does this mean? ). If published, this will include your full peer review and any attached files.

**Do you want your identity to be public for this peer review?** For information about this choice, including consent withdrawal, please see our Privacy Policy .

Reviewer #1: No

Reviewer #2: No

Reviewer #3: No

---

## [Decision Letter · Decision Letter 1]

26 Nov 2024

PGENETICS-D-24-00652R1RatXcan: A framework for cross-species integration of genome-wide association and gene expression dataPLOS GeneticsDear Dr. Im,

Thank you for submitting your manuscript to PLOS Genetics. After careful consideration, we feel that it has merit but does not fully meet PLOS Genetics's publication criteria as it currently stands. Therefore, we invite you to submit a revised version of the manuscript that addresses the points raised during the review process.

Please submit your revised manuscript within 30 days Dec 26 2024 11:59PM. If you will need more time than this to complete your revisions, please reply to this message or contact the journal office at plosgenetics@plos.org. Please include the following items when submitting your revised manuscript:

* A rebuttal letter that responds to each point raised by the editor and reviewer(s). You should upload this letter as a separate file labeled 'Response to Reviewers '. This file does not need to include responses to formatting updates and technical items listed in the 'Journal Requirements' section below. * A marked-up copy of your manuscript that highlights changes made to the original version. You should upload this as a separate file labeled 'Revised Manuscript with Track Changes '. * An unmarked version of your revised paper without tracked changes. You should upload this as a separate file labeled 'Manuscript '. If you would like to make changes to your financial disclosure, competing interests statement, or data availability statement, please make these updates within the submission form at the time of resubmission. Guidelines for resubmitting your figure files are available below the reviewer comments at the end of this letter.We look forward to receiving your revised manuscript.Kind regards,

Jingjing Yang, Ph.D.

Academic Editor

PLOS Genetics

Xiaofeng Zhu

Section Editor

PLOS Genetics

Aimée DudleyEditor-in-ChiefPLOS GeneticsAnne GorielyEditor-in-ChiefPLOS Genetics

**Additional Editor Comments:**

 Please address Reviewer2's remain comments.**Reviewers' comments:**

Reviewer's Responses to Questions

**Comments to the Authors:**

Reviewer #1: The authors have adequately addressed my previous comments.

Reviewer #2: The authors have appropriately addressed most of my previous comments. A few remaining minor issues:

1. There is a typo in the derivation of the covariance of \Gamma^(-1/2) (\mu+\epsilon) on page 17. The last line should be \Gamma^(-1/2) \sigma^2\Gamma \Gamma^(-1/2) = \sigma^2 I, and \sigma^2 is missing on both sides of the equation.

2. Now the authors have added type I error rates at the significance level of 1e-6 in Figure 3d, but the figure legend has not been updated.

3. The authors updated two p-values < 2.2e-16 in Figure 2 legend, but there are still three p-values showing P < 2.13e-30, P < 2.20e-16 and P < 2.20e-16 which I think are not very informative. Please show the small p-values with a better precision if possible.

Reviewer #3: My previous comments are all addressed.

**Have all data underlying the figures and results presented in the manuscript been provided?**

Reviewer #1: Yes

Reviewer #2: Yes

Reviewer #3: Yes

PLOS authors have the option to publish the peer review history of their article (what does this mean? ). If published, this will include your full peer review and any attached files.

**Do you want your identity to be public for this peer review?** For information about this choice, including consent withdrawal, please see our Privacy Policy .

Reviewer #1: No

Reviewer #2: No

Reviewer #3: No

**Figure resubmission:** While revising your submission, please upload your figure files to the Preflight Analysis and Conversion Engine (PACE) digital diagnostic tool, https://pacev2.apexcovantage.com/. PACE helps ensure that figures meet PLOS requirements. To use PACE, you must first register as a user. Registration is free. Then, login and navigate to the UPLOAD tab, where you will find detailed instructions on how to use the tool. If you encounter any issues or have any questions when using PACE, please email PLOS at figures@plos.org. Please note that Supporting Information files do not need this step. If there are other versions of figure files still present in your submission file inventory at resubmission, please replace them with the PACE-processed versions.**Reproducibility:** To enhance the reproducibility of your results, we recommend that authors deposit laboratory protocols in protocols.io, where a protocol can be assigned its own identifier (DOI) such that it can be cited independently in the future. Additionally, PLOS ONE offers an option to publish peer-reviewed clinical study protocols. Read more information on sharing protocols at https://plos.org/protocols?utm_medium=editorial-email&utm_source=authorletters&utm_campaign=protocols

---

## [Editor Report · Decision Letter 2]

20 Jan 2025

Dear Dr Im,

We are pleased to inform you that your manuscript entitled "RatXcan: A framework for cross-species integration of genome-wide association and gene expression data" has been editorially accepted for publication in PLOS Genetics. Congratulations!

Yours sincerely,

Jingjing Yang, Ph.D.

Academic Editor

PLOS Genetics

Xiaofeng Zhu

Section Editor

PLOS Genetics

Aimée Dudley

Editor-in-Chief

PLOS Genetics

Anne Goriely

Editor-in-Chief

PLOS Genetics

Comments from the reviewers (if applicable):

**Data Deposition**

http://datadryad.org/submit?journalID=pgenetics&manu=PGENETICS-D-24-00652R2

**Press Queries**

---

## [Editor Report · Acceptance letter]

PGENETICS-D-24-00652R2

RatXcan: A framework for cross-species integration of genome-wide association and gene expression data

Dear Dr Im,

We are pleased to inform you that your manuscript entitled "RatXcan: A framework for cross-species integration of genome-wide association and gene expression data" has been formally accepted for publication in PLOS Genetics! Your manuscript is now with our production department and you will be notified of the publication date in due course.

With kind regards,

Anita Estes

PLOS Genetics

On behalf of:
